# Different Curve Shapes of Fasting Glucose and Various Obesity-Related Indices by Diabetes and Sex

**DOI:** 10.3390/ijerph18063096

**Published:** 2021-03-17

**Authors:** Wei-Lun Wen, Pei-Yu Wu, Jiun-Chi Huang, Hung-Pin Tu, Szu-Chia Chen

**Affiliations:** 1Department of Internal Medicine, Kaohsiung Municipal Siaogang Hospital, Kaohsiung Medical University, Kaohsiung 812, Taiwan; stevenwen760829@gmail.com (W.-L.W.); wpuw17@gmail.com (P.-Y.W.); karajan77@gmail.com (J.-C.H.); 2Division of Endocrinology and Metabolism, Department of Internal Medicine, Kaohsiung Medical University Hospital, Kaohsiung Medical University, Kaohsiung 807, Taiwan; 3Division of Nephrology, Department of Internal Medicine, Kaohsiung Medical University Hospital, Kaohsiung Medical University, Kaohsiung 807, Taiwan; 4Faculty of Medicine, College of Medicine, Kaohsiung Medical University, Kaohsiung 807, Taiwan; 5Department of Public Health and Environmental Medicine, School of Medicine, College of Medicine, Kaohsiung Medical University, Kaohsiung 807, Taiwan; 6Research Center for Environmental Medicine, Kaohsiung Medical University, Kaohsiung 807, Taiwan

**Keywords:** fasting glucose, obesity related indices, diabetes, non-diabetes, sex difference

## Abstract

Fasting plasma glucose (FPG) and obesity-related indices are prognostic factors for adverse outcomes in both subjects with and without diabetes. A few studies have investigated sex differences in obesity indices related to the risk of diabetes, however no studies have compared the relationship between FPG and obesity-related indices by diabetes and sex. Therefore, in this study, we compared the curve shapes of FPG and various obesity-related indices by diabetes, and further explored sex differences in these associations. Data were derived from the Taiwan Biobank database, which included 5000 registered individuals. We used an adjusted generalized linear regression model and calculated the difference of least square means (Lsmean; standard error, SE) for males and females with and without diabetes. Associations between obesity-related indices and fasting glucose level by diabetes and sex groups were estimated, and the ORTHOREG procedure was used to construct B-splines. The post-fitting for linear models procedure was used to determine the range at which the trends separated significantly. The diabetes/sex/FPG interaction term was significant for all obesity-related indices, including body mass index, waist circumference, hip circumference, waist-to-hip ratio, waist-to-height ratio, lipid accumulation product, body roundness index, conicity index, body adiposity index and abdominal volume index. B-spline comparisons between males and females did not reach significance. However, FPG affected the trend towards obesity-related indices. As the fasting glucose level increased, the values of obesity-related indices varied more obviously in the participants without diabetes than in those with diabetes mellitus. The current study revealed that there was a different relationship between FPG and obesity-related indices by diabetes and sex. FPG affected the trend towards obesity-related indices more obviously in participants without diabetes than in those with diabetes. Further studies with a longitudinal design would provide a better understanding of the underlying mechanisms for the relationships.

## 1. Introduction

Diabetes is a serious health issue with an increasing prevalence worldwide. A recent study estimated a global prevalence of around about 9%, meaning that around 500 million people have diabetes worldwide. Moreover, the number is expected to increase by 25% in the next 10 years and 50% in the next 25 years [1]. The increase in patients with diabetes has spurred research efforts into disease control, and the incidence of acute complications such as cardiovascular disease (CVD), lower-extremity amputation, and all-cause mortality among people with diabetes has generally declined [2,3]. A modifiable risk factor for these complications is hyperglycemia, of which the underlying pathophysiology may be related to the induction of proinflammatory and prothrombotic pathways [4,5,6]. The American Diabetes Association guidelines recommended a treatment goal for fasting plasma glucose (FPG) that correlates with a glycated hemoglobin (HbA_1C_) level of <7% [7]. FPG is an important determinant of morbidity and mortality in patients with diabetes, and many studies have reported that FPG level is associated with microvascular complications, CVD, and mortality in both people with prediabetes and those without diabetes [8,9,10,11,12]. Furthermore, the level of FPG has been associated with the risk of future progression to diabetes in this population [13,14,15,16,17].

Obesity is the major etiological cause and clinical manifestation of diabetes, and the term “diabesity” has been proposed [18]. Obesity is also a known risk factor for many other diseases, including metabolic syndrome, dyslipidemia, hypertension, CVD, fatty liver diseases, musculoskeletal diseases, and even cancer [19,20,21,22,23]. Therefore, obesity causes a huge health economic burden [24]. Obesity can be classified as general and central obesity, with body mass index (BMI) commonly used to measure general obesity [25] and waist circumference (WC) to measure central obesity [19]. Apart from a single direct WC measurement, several more complex anthropometric indices have been developed to define and quantify central obesity, including waist-to-hip ratio (WHR), waist-to-height ratio (WHtR), lipid accumulation product (LAP), body roundness index (BRI), conicity Index (CI), body adiposity index (BAI) and abdominal volume index (AVI). Many studies have confirmed the associations among these obesity-related indices and the risk of diabetes [26,27,28,29,30]. However, only a few studies have revealed obvious sex differences. Several studies have reported that some indices are adequate to discriminate diabetes from metabolically healthy individuals in both sexes, whereas other prospective studies have found a difference in some indices for the prediction of the future risk of diabetes progression [31,32,33]. The results of these studies have been inconsistent. Moreover, no studies have compared the relationship between FPG and obesity-related indices by diabetes and sex.

Though the sex difference of metabolism was determined in diabetes mellitus (DM) or prediabetes population [26,27,28,29,30], whether this difference could extend to non-DM population was uncertain, and this could affect the screening policy of related metabolic diseases in different sexes. Therefore, in this study, we enrolled 5000 individuals from the Taiwan Biobank (TWB) database and compared curve shapes of FPG and various obesity-related indices by diabetes, and further explored sex differences in these associations.

## 2. Materials and Methods

### 2.1. The Taiwan Biobank

The TWB is the largest government-supported biobank in Taiwan [34,35]. It includes genomic and lifestyle data of community-based individuals aged 30–70 years with no history of cancer, with all registered individuals providing informed consent, blood samples, and information on personal and lifestyle factors through questionnaires administered by TWB researchers. All of the individuals also underwent physical examinations. In this study, we included 5000 individuals registered in the TWB from December 2008 to April 2014.

### 2.2. Collection of Demographic, Medical and Laboratory Data

The TWB includes data on body weight, height, WC, hip circumference (HC), WHR, WHtR and BMI. The following baseline variables were recorded: demographic features (age and sex), medical history (DM), history of smoking and drinking alcohol, examination findings (systolic blood pressure (SBP) and diastolic blood pressure (DBP)) and laboratory data (estimated glomerular filtration rate (eGFR) and uric acid, fasting glucose, HbA_1c_, total cholesterol, low-density lipoprotein (LDL) cholesterol, high-density lipoprotein (HDL) cholesterol, and triglycerides (TG)). After ten minutes rest, the blood pressures were averaged from three times measurement. Fasting blood samples were obtained, and laboratory data were measured using an autoanalyzer (Roche Diagnostics GmbH, D-68298 Mannheim COBAS Integra 400). Serum creatinine was measured according to the compensated Jaffé (kinetic alkaline picrate) method using the same autoanalyzer (Roche/Integra 400, Roche Diagnostics) and a calibrator that could be used in isotope-dilution mass spectrometry. The eGFR was calculated using the Modification of Diet in Renal Disease 4-variable equation [36].

### 2.3. Definitions of Diabetes and Non-Diabetes

Participates who had a past history of diabetes, used hypoglycemic agents, and had a fasting glucose level ≥126 mg/dL or HbA_1c_ ≥ 6.5% were considered to have diabetes (diabetes group) [37]. Participates who had no past history of diabetes and whose fasting glucose level was <126 mg/dL and HbA_1c_ was <6.5% were considered to not have diabetes (non-diabetes group).

### 2.4. Obesity-Related Indices

For males, LAP was calculated as: LAP = $\left({\mathrm{WC}}_{(\mathrm{cm})\;}-\;65\right){\;\times \;\mathrm{TG}}_{(\mathrm{mmol}/L)}$; and for females:

LAP = $\left({\mathrm{WC}}_{\left(\mathrm{cm}\right)}\;-\;58\right){\;\times \;\mathrm{TG}}_{(\mathrm{mmol}/L)}$ [38].

BRI was calculated as: BRI = $364.2\;-\;365.5\;\times \;\sqrt{1\;-\;{\left(\frac{\frac{{\mathrm{WC}}_{\left(m\right)}}{2\pi }}{0.5{\;\times \;\mathrm{BH}}_{\left(m\right)}}\right)}^{2}}$ [39].

CI was calculated using the Valdez equation based on the values for body weight, height and WC: CI = $\frac{{\mathrm{WC}}_{\left(m\right)}}{0.109\;\times \;\sqrt{\frac{{\mathrm{BW}}_{\left(\mathrm{kg}\right)}}{{\mathrm{BH}}_{\left(m\right)}}}}$ [40].

BAI was calculated according to the method of Bergman and colleagues as:

BAI = $\frac{{\;\mathrm{Hip}\;\mathrm{circumference}}_{\left(\mathrm{cm}\right)\;}}{{\mathrm{BH}}_{\left(m\right)}{}^{3/2}}\;-18$ [41].

AVI was calculated as: AVI = $\frac{{2\;\times \;(\mathrm{WC}}_{\left(\mathrm{cm}\right)}{)}^{2}\;+\;0.7{\;\times \;(\mathrm{WC}}_{\left(\mathrm{cm}\right)}\;{-\;\mathrm{HC}}_{\left(\mathrm{cm}\right)}{)}^{2}}{1000}$ [42]. 

### 2.5. Ethics Statement

Ethical approval was granted by the Ethics and Governance Council of the TWB and the Institutional Review Board on Biomedical Science Research/IRB-BM, Academia Sinica, Taiwan. Each participant provided written informed consent, and the study was conducted in accordance with the principles of the Declaration of Helsinki. In addition, the Institutional Review Board of Kaohsiung Medical University Hospital approved this study (KMUHIRB-E(I)-20180242).

### 2.6. Statistical Analysis

Continuous variables are presented as mean (standard deviation, SD). Continuous outcome variables exhibiting a skewed distribution were transformed using the natural logarithms. Categorical variables are expressed as number of subjects (%). Data of continuous and categorical variables were analyzed using the t test and chi-squared test to compare the diabetes group and with that of the comparison group according to sex.

Next, we conducted the adjusted generalized linear regression model and calculated the difference of least square means (Lsmean; standard error, SE) for males and females with and without DM. Multiple comparison analysis testing was by using Bonferroni method. The interaction between DM and sex was tested after adjusted covariates were included such as DBP, total cholesterol, Ln (TG), HDL-cholesterol, eGFR and uric acid.

Outcome variables included were BMI, WC, HC, WHR, WHtR, LAP, BRI, CI, BAI and AVI. Outcome variables in fasting glucose level according to DM by sex groups were estimated and the ORTHOREG procedure constructs B-splines. To perform multiple comparisons among predicted values in a model with group-specific trends (DM and sex) that are modeled through regression splines. In order to determine the range on which the trends separate significantly, the post-fitting for linear models (PLM) procedure is executed.

All data was analyzed using Statistical Analysis Software, version 9.4 (SAS Institute, Cary, NC, USA) with a statistically significant level of two tailed *p*-value < 0.05.

## 3. Results 

The mean age of the 5000 participants (2335 males and 2665 females) was 49.6 ± 10.7 years. The overall prevalence rate of type 2 DM was 10.3%. The participants were stratified into four groups according to DM and sex as follows: DM males (*n* = 295), non-DM males (*n* = 2040), DM females (*n* = 220) and non-DM females (*n* = 2445).

### 3.1. Comparison of Clinical Characteristics of the Study Population between Males and Females with and without DM

A comparison of the clinical characteristics among the participants with and without DM indifferent sex is shown in Table 1. In males, compared to the participants without DM, those with DM were older, had higher weight, lower height, higher BMI, higher WC, higher HC, higher WHR, higher WHtR, higher SBP, higher DBP, higher fasting glucose, higher HbA_1c_, higher TG, lower total cholesterol, lower HDL-cholesterol, lower LDL-cholesterol, higher eGFR and lower uric acid. Regarding obesity-related indices, in males, compared to the participants without DM, those with DM had higher LAP, higher BRI, higher CI, higher VAI, higher BAI and higher AVI. In females, compared to the participants without DM, those with DM were older, had higher weight, lower height, higher BMI, higher WC, higher HC, higher WHR, higher WHtR, higher SBP, higher DBP, higher fasting glucose, higher HbA_1c_, higher TG, higher total cholesterol, lower HDL-cholesterol, higher LDL-cholesterol, higher uric acid, higher LAP, higher BRI, higher CI, higher VAI, higher BAI and higher AVI.

### 3.2. B-Spline Comparisons for Fasting Glucose with Obesity-Related Indices

Multiple comparison analysis testing of the interaction between DM and sex after adjustments for age, DBP, total cholesterol, Ln (TG), HDL-cholesterol, eGFR and uric acid is shown in Table 2. Whether in non-DM or DM participants, B-spline comparisons between males and females are not achieving significance. In males, comparing DM and non-DM participants, B-spline comparisons achieved significance in WHR, LAP and CI. In females, comparing DM and non-DM participants, B-spline comparisons achieved significance in LAP and BAI. However, the DM–Sex–fasting glucose interaction term was calculated significantly for all obesity-related indices, including BMI, WC, HC, WHR, WHtR, LAP, BRI, CI, BAI and AVI.

### 3.3. The Relationship between Fasting Glucose and Various Obesity-Related Indices by DM and Sex

Figure 1 illustrates the relationship between fasting glucose and various obesity-related indices by DM and sex: BMI (A), WC (B), HC(C), WHR (D), WHtR (E), LAP (F), BRI (G), CI (H), BAI (I) and AVI (J). The figure presents the sex difference of obesity-related indices in association with continuous FPG change more directly in non-diabetic participants.

### 3.4. Separate Trends between Obesity-Related Indices and Fasting Glucose between Males and Females with and without DM

In order to determine the range on which the trends separate significantly, the PLM procedure was executed (Table 3). In Table 3, we grabbed part of the range of fasting glucose: 70–110 mg/dL in non-DM and 120–160 mg/dL in DM. Although B-spline comparisons between males and females did not achieve significance, whether DM or non-DM in Table 2, however, fasting glucose affected the trend towards obesity-related indices. While fasting glucose increased, the values of obesity-related indices varied more obvious in non-DM than in DM participants.

## 4. Discussion

In this study, the different correspondences between obesity-related indices and DM status in different sexes were unmasked if we took FPG into consideration. Furthermore, the differences were more obvious in the non-DM group than in the DM group, but gradually declined as the FPG increased in the general Taiwanese population.

The first important finding of this study is that there were differences in the relationships between FPG and obesity-related indices by DM and sex. Sex differences in obesity indices related to the risk of diabetes have been reported in previous studies [31,32,33]. The underlying mechanism could be complex. In addition to the persistent biological effects of different sex hormones and sex-specific gene expressions, lifelong psychosocial factors such as gender-sensitive economic, behavioral, cultural, and environmental factors may also aggravate the difference between males and females, and the overall reason may be explained as evolutionary maladaptation to relative food security in the modern age [43,44]. Previous studies have reported that males have larger increases in WC with weight gain than women, the so called “apple shape” in males and “pear shape” in females [45,46], and this may partially explain why the prevalence of diabetes or FPG is mildly higher in males than in females [47,48,49].

The second important finding of this study is that B-spline comparisons between males and females were not significant in the non-DM group. However, we found more obvious sex differences in the association between obesity-related indices and continuous changes in FPG in the non-diabetes group. Previous studies have reported inconclusive results of gender differences in indices related to the risk of diabetes. Although some studies have concluded that central obesity based on WC or WHR may be a more specific risk factor in males than in females [50,51], other studies have reported opposite conclusions, in that the general obesity index BMI is more specific in males and that central obesity indices are more specific in female [27,52,53]. In our study, the females in the non-DM group were more compatible with background knowledge, because a higher FPG level related to increasing obesity indices and could be regarded as a risk for diabetes [14,15]. In contrast, the obesity indices in males decreased to a nadir before FPG fell to the lowest point and then rebounded, resulting in a U-shaped curve, which has never been reported before. This implies that some overweight or obese males had very low FPG levels far from a diagnosis of diabetes, similar to the concept of metabolically healthy obesity, which likely represents a transient phenotype between lean men without diabetes and obese men with impaired fasting glucose. Therefore, at least avoiding further weight gain would be recommended in these individuals [54,55].

The other important finding of this study is that FPG affected the trend towards obesity-related indices more obviously in the non-DM group than in the DM group. Managing obesity is recommended in all obese or overweight diabetic patients as it can improve further glycemic control by improving insulin resistance [56,57]. In the current study, the obesity indices were basically stable and accompanied increasing FPG in the DM group, which is in contrast to some previous studies [58,59,60,61]. There are several reasons that could explain this discrepancy. First, due to the study design, we could not investigate the duration of diabetes as in previous studies, and this may have affected the results of glycemic burden independently of obesity status. Second, we did not have records of anti-hyperglycemic medication use in our participants, and this may also have impacted glycemic control. Third, rather than using HbA_1C_ to evaluate glycemic burden as in the prior studies, we used FPG. FPG shows greater variability than HbA_1C_ in patients with diabetes, and this could have amplified the influence of various factors, such as the aforementioned anti-hyperglycemic medication history.

Furthermore, we observed that differences in obesity-related indices corresponding to the same FPG level between sexes gradually decreased; that is, the two curves gradually converged, along with increasing FPG from the non-diabetic to diabetic participants. The reason why the associations among FPG, obesity-related indices, and sex did not extend from the non-diabetic to diabetic participants may be due to the aforementioned unknown duration of diabetes and anti-hyperglycemic medication use, as both can attenuate the impact of sex- and obesity-related indices. In addition, the nadir of the FPG level corresponded to a BMI of 22.5 kg/m^2^ and WC of 77 cm in the female participants, and a BMI of 24.5 kg/m^2^ and WC of 87 cm in the male participants. This observation generally responds to the different criteria for WC in men and women to diagnose metabolic syndrome, however previous same BMI definitions of overweight or obesity for males and females do not reflect different thresholds for metabolic abnormalities [25,62,63,64].

The strengths of the current study include detailed data collection from 5000 individuals, and the new finding of different curve shapes of FPG and various obesity-related indices by diabetes and sex, which has not previously been reported. There are two major limitations to this study. First, this was a cross-sectional study, so we could not define causal relationships between the obesity indices and FPG. Second, due to the study design, we did not have data on some important determinants of diabetes, including anti-hyperglycemic medications and the duration of diabetes.

## 5. Conclusions

In conclusion, the current study revealed a different trend of obesity-related indices in different sexes if we compared not only DM status, but also included the FPG. FPG affected the trend towards obesity-related indices more obviously in the non-DM group than in the DM group. Further studies with longitudinal designs would provide a better understanding of the underlying mechanisms for these relationships.

## Figures and Tables

**Figure 1 ijerph-18-03096-f001:**
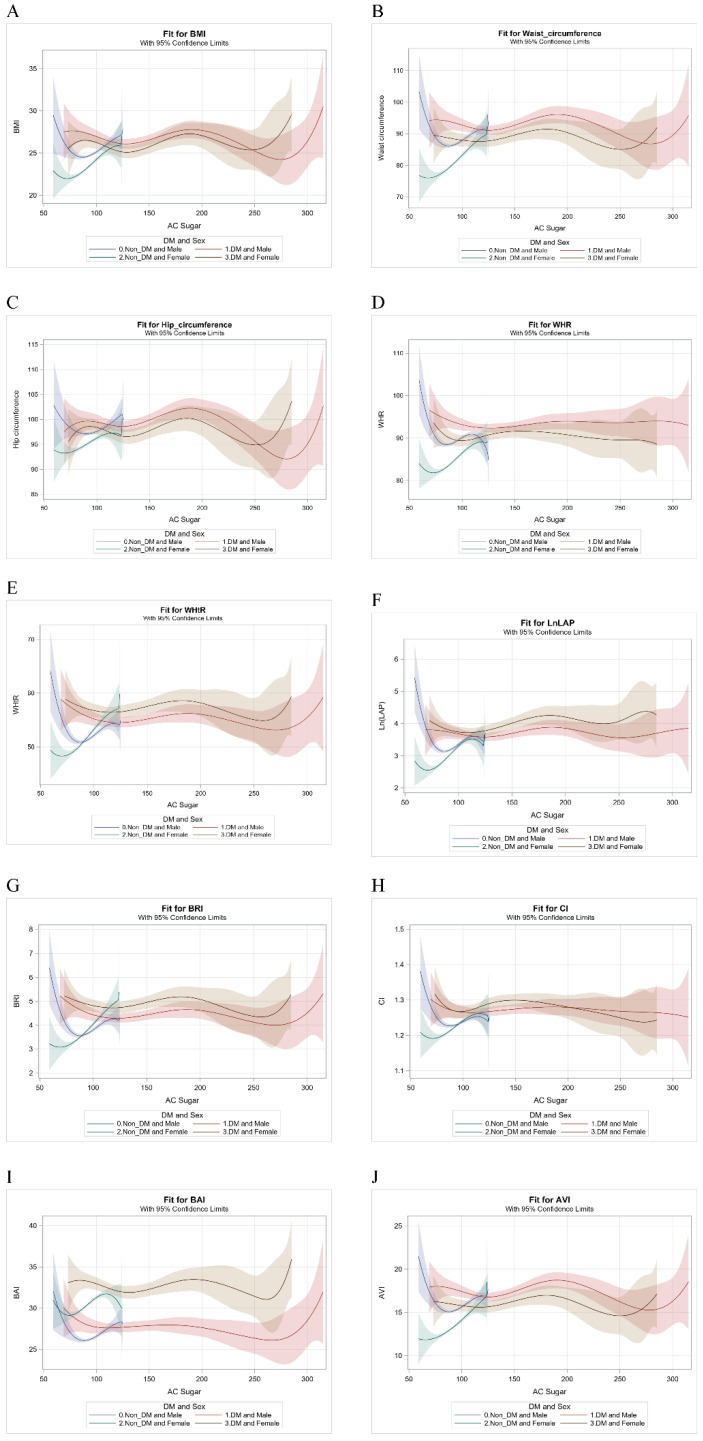
The relationship between fasting glucose (AC sugar) and various obesity-related indices by DM and sex: BMI (**A**), WC (**B**), HC (**C**), WHR (**D**), WHtR (**E**), LAP (**F**), BRI (**G**), CI (**H**), BAI (**I**) and AVI (**J**). Abbreviations. DM, diabetes mellitus; BMI, body mass index; WC, waist circumference; HC, hip circumference; WHR, waist-to-hip ratio; WHtR, waist-to-height ratio; LAP, lipid accumulation product; BRI, body roundness index; CI, conicity index; BAI, body adiposity index; AVI, abdominal volume index.

**Table 1 ijerph-18-03096-t001:** Characteristics of the study population between males and females with and without diabetes mellitus (DM).

	Males			Females		
	DM	Non-DM	*p*	DM	Non-DM	*p*
*N*	295	2040		220	2445	
Age (year)	55.80 (8.76)	48.74 (11.00)	<0.0001	56.38 (8.73)	48.91 (10.29)	<0.0001
Smoking status, *N* (%)						
Never + Occasional	127 (43.1)	985 (48.3)		206 (93.6)	2303 (94.2)	
Quit drinking	100 (33.9)	674 (33.0)		9 (4.1)	111 (4.5)	
Regular	68 (23.1)	381 (18.7)	0.1285	5 (2.3)	31 (1.3)	0.4481
Drinking status, *N* (%)						
Never + Occasional	226 (76.6)	1625 (79.7)		215 (97.7)	2395 (98.0)	
Quit drinking	24 (8.1)	97 (4.8)		3 (1.4)	13 (0.5)	
Regular	45 (15.3)	318 (15.6)	0.0497	2 (0.9)	37 (1.5)	0.2428
Weight (kg)	75.21 (12.91)	71.26 (10.62)	<0.0001	62.56 (10.20)	57.37 (8.55)	<0.0001
Height (cm)	167.84 (6.24)	169.08 (6.28)	0.0015	155.35 (5.50)	157.22 (5.50)	<0.0001
BMI (kg/m^2^)	26.61 (3.80)	24.89 (3.15)	<0.0001	25.92 (4.00)	23.22 (3.30)	<0.0001
WC (cm)	92.41 (9.24)	87.14 (8.33)	<0.0001	88.48 (9.82)	80.26 (8.85)	<0.0001
HC (cm)	99.35 (7.16)	97.47 (6.22)	<0.0001	97.79 (7.57)	95.08 (6.39)	<0.0001
WHR	0.93 (0.05)	0.89 (0.05)	<0.0001	0.90 (0.07)	0.84 (0.06)	<0.0001
WHtR	0.55 (0.05)	0.52 (0.05)	<0.0001	0.57 (0.07)	0.51 (0.06)	<0.0001
SBP (mm Hg)	126.19 (16.34)	118.32 (15.48)	<0.0001	124.9 (17.29)	110.9 (17.03)	<0.0001
DBP (mm Hg)	76.59 (10.67)	75.22 (10.65)	0.0400	72.68 (10.06)	67.47 (10.20)	<0.0001
Laboratory parameters						
AC sugar (mg/dL)	137.85 (41.67)	94.43 (7.55)	<0.0001	129.89 (39.53)	90.5 (7.44)	<0.0001
HbA_1c_ (%)	7.55 (1.46)	5.61 (0.32)	<0.0001	7.48 (1.43)	5.57 (0.34)	<0.0001
Triglyceride (mg/dL)	161.08 (112.15)	132.21 (99.77)	<0.0001	175.45 (145.61)	97.31 (60.89)	<0.0001
Total cholesterol (mg/dL)	185.23 (37.65)	194.13 (34.56)	<0.0001	204.20 (40.33)	196.07 (36.25)	0.0016
HDL-cholesterol (mg/dL)	44.79 (9.93)	49.66 (11.41)	<0.0001	51.18 (10.57)	59.47 (13.43)	<0.0001
LDL-cholesterol (mg/dL)	115.78 (32.88)	124.5 (31.95)	<0.0001	125.95 (36.85)	120.49 (32.06)	0.0170
eGFR (mL/min/1.73 m^2^)	70.43 (18.49)	68.13 (13.78)	0.0107	111.92 (30.15)	110.88 (24.51)	0.5543
Uric acid (mg/dL)	6.23 (1.55)	6.51 (1.38)	0.0015	5.50 (1.35)	4.83 (1.09)	<0.0001
Obesity-related indices						
LAP	53.26 (48.59)	35.38 (36.01)	<0.0001	61.39 (60.2)	26.16 (22.48)	<0.0001
BRI	4.41 (1.10)	3.71 (0.97)	<0.0001	4.85 (1.47)	3.64 (1.20)	<0.0001
CI	1.27 (0.06)	1.23 (0.06)	<0.0001	1.28 (0.08)	1.22 (0.08)	<0.0001
BAI	27.73 (3.29)	26.39 (3.0)	<0.0001	32.59 (4.37)	30.31 (3.68)	<0.0001
AVI	17.3 (3.48)	15.42 (2.92)	<0.0001	15.94 (3.58)	13.22 (2.87)	<0.0001

Data are presented as mean (standard deviation, SD) or number of subjects (%). Data for triglycerides and LAP was skewed and log transformed for analysis. Abbreviations. DM, diabetes mellitus; BMI, body mass index; WC, waist circumference; HC, hip circumference; WHR, waist-to-hip ratio; WHtR, waist-to-height ratio; SBP, systolic blood pressure; DBP, diastolic blood pressure; AC sugar, fasting glucose; HDL, high-density lipoprotein; LDL, low-density lipoprotein; eGFR, estimated glomerular filtration rate; LAP, lipid accumulation product; BRI, body roundness index; CI, conicity index; BAI, body adiposity index; AVI, abdominal volume index. The differences between groups were checked by Chi-square test for categorical variables and by independent t-test for continuous variables.

**Table 2 ijerph-18-03096-t002:** B-spline comparisons for DM by sex or sex by DM in AC sugar levels with obesity-related indices.

	Non-DM		DM		Males		Female		DM by Sex by AC Sugar *	
	Males vs. Females		Males vs. Females		Non-DM vs. DM		Non-DM vs. DM			
	β (SE)	*p*	β (SE)	*p*	β (SE)	*p*	β (SE)	*p*	β (SE)	*p* for Interaction
BMI (kg/m^2^) ^†^	9.22 (27.92)	0.7412	−35.92 (238.92)	0.8805	7.23 (16.55)	0.6624	−16.75 (22.58)	0.4583	−0.03 (0.01)	0.0123
WC (cm) ^†^	−38.53 (73.39)	0.5996	−77.14 (628.05)	0.9023	57.38 (43.51)	0.1874	−14.46 (59.36)	0.8076	−0.11 (0.03)	0.0012
HC (cm) ^†^	45.46 (54.75)	0.4064	−147.34 (468.58)	0.7532	−8.67 (32.46)	0.7894	−36.14 (44.29)	0.4145	−0.05 (0.03)	0.0473
WHR *100 ^†^	−74.23 (50.24)	0.1396	51.93 (429.97)	0.9039	120.49 (29.79)	<0.0001	−42.91 (40.64)	0.2911	−0.07 (0.02)	0.0028
WHtR *100 ^†^	−28.70 (46.50)	0.5372	−136.49 (397.92)	0.7316	48.17 (27.57)	0.0806	−22.49 (37.61)	0.5500	−0.08 (0.02)	0.0005
LnLAP ^§^	−1.13 (6.45)	0.8612	22.51 (55.14)	0.6831	11.37 (3.83)	0.0030	−10.70 (5.21)	0.0402	−0.01 (0.00)	0.0380
BRI ^†^	−9.11 (9.38)	0.3317	−25.14 (80.28)	0.7542	9.58 (5.56)	0.0851	−1.18 (7.59)	0.8761	−0.02 (0.00)	0.0004
CI ^†^	−0.23 (0.64)	0.7189	−0.95 (5.44)	0.8619	0.76 (0.38)	0.0443	−0.53 (0.51)	0.3054	0.00 (0.00)	0.0059
BAI ^†^	36.5 (29.07)	0.2093	−204.4 (248.80)	0.4114	8.27 (17.24)	0.6316	−51.35 (23.52)	0.0290	−0.03 (0.01)	0.0498
AVI ^†^	−20.58 (25.01)	0.4107	−22.70 (214.04)	0.9155	18.73 (14.83)	0.2067	3.62 (20.23)	0.8581	−0.03 (0.01)	0.0042

SE: standard error. Abbreviations. DM, diabetes mellitus; BMI, body mass index; WC, waist circumference; HC, hip circumference; WHR, waist-to-hip ratio; WHtR, waist-to-height ratio; LAP, lipid accumulation product; BRI, body roundness index; CI, conicity index; BAI, body adiposity index; AVI, abdominal volume index. The ORTHOREG procedure constructs B-splines. Group-specific trends (DM and sex) that are modeled through regression splines. * The DM–sex–AC sugar interaction term was estimated after adjusted for age, DBP, total cholesterol, Ln (triglyceride), HDL-cholesterol, eGFR and uric acid variables by using a generalized linear model. ^†^ Adjusted for age, DBP, DBP, total cholesterol, Ln (triglyceride), HDL-cholesterol, eGFR and uric acid variables. ^§^ Adjusted for age, DBP, total cholesterol, HDL-cholesterol, eGFR and uric acid variables. Data for triglycerides and LAP was skewed and log transformed for analysis.

**Table 3 ijerph-18-03096-t003:** Separate trends between obesity-related indices and fasting glucose between males and females with and without DM.

	Males	Females	Males vs. Females			Male	Female	Males vs. Females	
Non-DM	Lsmean (SE)	Lsmean (SE)	β (SE) /Difference of Lsmean (95% CI)	*p*	DM	Lsmean (SE)	Lsmean (SE)	β (SE) /Difference of Lsmean (95% CI)	*p*
BMI									
At AC_Sugar = 70	26.13 (0.83)	22.00 (0.5)	4.13 (2.24, 6.03)	0.0001	At AC_Sugar = 120	26.17 (0.26)	25.25 (0.33)	0.92 (0.10, 1.74)	0.1713
At AC_Sugar = 80	24.75 (0.24)	22.23 (0.13)	2.53 (1.99, 3.06)	<0.0001	At AC_Sugar = 130	26.08 (0.30)	25.10 (0.38)	0.98 (0.03, 1.93)	0.0428
At AC_Sugar = 90	24.56 (0.09)	23.10 (0.08)	1.46 (1.22, 1.69)	<0.0001	At AC_Sugar = 140	26.24 (0.31)	25.31 (0.38)	0.94 (−0.03, 1.90)	0.0571
At AC_Sugar = 100	25.13 (0.11)	24.24 (0.12)	0.88 (0.56, 1.20)	<0.0001	At AC_Sugar = 150	26.57 (0.31)	25.74 (0.40)	0.83 (−0.16, 1.82)	0.1005
At AC_Sugar = 110	26.05 (0.19)	25.30 (0.29)	0.75 (0.07, 1.43)	0.1843	At AC_Sugar = 160	26.97 (0.36)	26.27 (0.46)	0.69 (−0.45, 1.84)	0.2359
WC									
At AC_Sugar = 70	91.68 (2.17)	76.05 (1.33)	15.63 (10.64, 20.62)	<0.0001	At AC_Sugar = 120	91.16 (0.69)	87.59 (0.86)	3.56 (1.41, 5.72)	0.0072
At AC_Sugar = 80	86.94 (0.63)	77.35 (0.35)	9.59 (8.18, 11.00)	<0.0001	At AC_Sugar = 130	91.15 (0.79)	87.99 (0.99)	3.16 (0.67, 5.65)	0.0128
At AC_Sugar = 90	86.23 (0.24)	79.95 (0.20)	6.28 (5.66, 6.90)	<0.0001	At AC_Sugar = 140	91.74 (0.81)	88.72 (1.01)	3.02 (0.48, 5.56)	0.0197
At AC_Sugar = 100	87.86 (0.28)	83.2 (0.32)	4.65 (3.81, 5.50)	<0.0001	At AC_Sugar = 150	92.71 (0.82)	89.61 (1.04)	3.10 (0.50, 5.70)	0.0196
At AC_Sugar = 110	90.15 (0.51)	86.49 (0.76)	3.66 (1.87, 5.45)	0.0004	At AC_Sugar = 160	93.85 (0.94)	90.48 (1.22)	3.36 (0.34, 6.38)	0.0290
HC									
At AC_Sugar = 70	99.31 (1.62)	93.26 (0.99)	6.05 (2.32, 9.77)	0.0088	At AC_Sugar = 120	98.64 (0.52)	96.82 (0.64)	1.82 (0.21, 3.43)	0.1595
At AC_Sugar = 80	97.61 (0.47)	93.82 (0.26)	3.79 (2.74, 4.84)	<0.0001	At AC_Sugar = 130	98.67 (0.59)	96.57 (0.74)	2.10 (0.24, 3.96)	0.0268
At AC_Sugar = 90	97.13 (0.18)	94.99 (0.15)	2.13 (1.67, 2.59)	<0.0001	At AC_Sugar = 140	99.15 (0.60)	96.96 (0.75)	2.18 (0.29, 4.08)	0.0239
At AC_Sugar = 100	97.60 (0.21)	96.27 (0.24)	1.33 (0.69, 1.96)	0.0002	At AC_Sugar = 150	99.89 (0.61)	97.75 (0.78)	2.14 (0.20, 4.08)	0.0308
At AC_Sugar = 110	98.77 (0.38)	97.15 (0.57)	1.62 (0.29, 2.96)	0.1036	At AC_Sugar = 160	100.73 (0.70)	98.69 (0.91)	2.04 (−0.21, 4.29)	0.0760
WHR*100									
At AC_Sugar = 70	92.68 (1.49)	81.87 (0.91)	10.82 (7.40, 14.23)	<0.0001	At AC_Sugar = 120	92.38 (0.47)	90.40 (0.59)	1.98 (0.51, 3.46)	0.0508
At AC_Sugar = 80	88.82 (0.43)	82.30 (0.24)	6.52 (5.55, 7.48)	<0.0001	At AC_Sugar = 130	92.35 (0.54)	91.01 (0.68)	1.34 (−0.36, 3.05)	0.1225
At AC_Sugar = 90	88.68 (0.16)	84.13 (0.14)	4.54 (4.12, 4.97)	<0.0001	At AC_Sugar = 140	92.49 (0.55)	91.39 (0.69)	1.10 (−0.64, 2.84)	0.2148
At AC_Sugar = 100	90.10 (0.19)	86.47 (0.22)	3.63 (3.05, 4.21)	<0.0001	At AC_Sugar = 150	92.76 (0.56)	91.60 (0.71)	1.17 (−0.61, 2.95)	0.1984
At AC_Sugar = 110	90.91 (0.35)	88.41 (0.52)	2.49 (1.27, 3.72)	0.0004	At AC_Sugar = 160	93.10 (0.64)	91.64 (0.84)	1.46 (−0.60, 3.53)	0.1651
WHtR*100									
At AC_Sugar = 70	55.06 (1.38)	48.27 (0.84)	6.79 (3.63, 9.95)	0.0002	At AC_Sugar = 120	54.51 (0.44)	56.43 (0.54)	−1.92 (−3.29, −0.56)	0.0351
At AC_Sugar = 80	51.44 (0.40)	49.03 (0.22)	2.41 (1.51, 3.30)	<0.0001	At AC_Sugar = 130	54.56 (0.50)	56.65 (0.63)	−2.08 (−3.66, −0.50)	0.0098
At AC_Sugar = 90	50.91 (0.15)	50.90 (0.13)	0.01 (−0.38, 0.40)	1.0000	At AC_Sugar = 140	54.80 (0.51)	57.03 (0.64)	−2.23 (−3.84, −0.62)	0.0066
At AC_Sugar = 100	52.12 (0.18)	53.24 (0.21)	−1.12 (−1.66, −0.59)	0.0002	At AC_Sugar = 150	55.16 (0.52)	57.51 (0.66)	−2.35 (−4.00, −0.71)	0.0051
At AC_Sugar = 110	53.70 (0.32)	55.45 (0.48)	−1.74 (−2.87, −0.61)	0.0157	At AC_Sugar = 160	55.55 (0.59)	57.99 (0.77)	−2.44 (−4.35, −0.53)	0.0124
LnLAP									
At AC_Sugar = 70	3.83 (0.19)	2.56 (0.12)	1.28 (0.83, 1.72)	<0.0001	At AC_Sugar = 120	3.58 (0.06)	3.75 (0.08)	−0.16 (−0.35, 0.03)	0.5616
At AC_Sugar = 80	3.23 (0.06)	2.67 (0.03)	0.56 (0.44, 0.69)	<0.0001	At AC_Sugar = 130	3.59 (0.07)	3.81 (0.09)	−0.23 (−0.44, −0.01)	0.0433
At AC_Sugar = 90	3.16 (0.02)	2.97 (0.02)	0.19 (0.13, 0.24)	<0.0001	At AC_Sugar = 140	3.63 (0.07)	3.91 (0.09)	−0.28 (−0.50, −0.06)	0.0144
At AC_Sugar = 100	3.36 (0.03)	3.30 (0.03)	0.06 (−0.02, 0.13)	0.8373	At AC_Sugar = 150	3.69 (0.07)	4.01 (0.09)	−0.32 (−0.55, −0.09)	0.0061
At AC_Sugar = 110	3.57 (0.04)	3.51 (0.07)	0.06 (−0.09, 0.22)	1.0000	At AC_Sugar = 160	3.76 (0.08)	4.11 (0.11)	−0.35 (−0.61, −0.08)	0.0098
BRI									
At AC_Sugar = 70	4.48 (0.28)	3.08 (0.17)	1.40 (0.76, 2.03)	0.0001	At AC_Sugar = 120	4.29 (0.09)	4.73 (0.11)	−0.44 (−0.71, −0.16)	0.0116
At AC_Sugar = 80	3.71 (0.08)	3.24 (0.04)	0.47 (0.28, 0.65)	<0.0001	At AC_Sugar = 130	4.30 (0.10)	4.77 (0.13)	−0.46 (−0.78, −0.14)	0.0045
At AC_Sugar = 90	3.58 (0.03)	3.59 (0.03)	−0.02 (−0.10, 0.06)	1.0000	At AC_Sugar = 140	4.36 (0.10)	4.85 (0.13)	−0.49 (−0.82, −0.17)	0.0030
At AC_Sugar = 100	3.81 (0.04)	4.05 (0.04)	−0.24 (−0.35, −0.13)	<0.0001	At AC_Sugar = 150	4.43 (0.10)	4.95 (0.13)	−0.52 (−0.85, −0.19)	0.0022
At AC_Sugar = 110	4.13 (0.06)	4.52 (0.10)	−0.40 (−0.63, −0.17)	0.0039	At AC_Sugar = 160	4.51 (0.12)	5.06 (0.16)	−0.54 (−0.93, −0.16)	0.0059
CI									
At AC_Sugar = 70	1.28 (0.02)	1.19 (0.01)	0.08 (0.04, 0.13)	0.0008	At AC_Sugar = 120	1.27 (0.01)	1.29 (0.01)	−0.02 (−0.04, 0.00)	0.2744
At AC_Sugar = 80	1.23 (0.01)	1.20 (0.00)	0.04 (0.02, 0.05)	<0.0001	At AC_Sugar = 130	1.27 (0.01)	1.29 (0.01)	−0.02 (−0.05, 0.00)	0.0239
At AC_Sugar = 90	1.23 (0.00)	1.22 (0.00)	0.01 (0.00, 0.01)	0.0107	At AC_Sugar = 140	1.27 (0.01)	1.30 (0.01)	−0.03 (−0.05, 0.00)	0.0182
At AC_Sugar = 100	1.24 (0.00)	1.24 (0.00)	0.00(−0.01, 0.00)	1.0000	At AC_Sugar = 150	1.27 (0.01)	1.30 (0.01)	−0.02 (−0.05, 0.00)	0.0307
At AC_Sugar = 110	1.25 (0.00)	1.26 (0.01)	−0.01 (−0.02, 0.01)	1.0000	At AC_Sugar = 160	1.28 (0.01)	1.30 (0.01)	−0.02 (−0.05, 0.01)	0.1173
BAI									
At AC_Sugar = 70	28.10 (0.86)	29.24 (0.53)	−1.14 (−3.12, 0.84)	1.0000	At AC_Sugar = 120	27.63 (0.27)	32.08 (0.34)	−4.45 (−5.30, −3.59)	<0.0001
At AC_Sugar = 80	26.42 (0.25)	29.32 (0.14)	−2.90 (−3.45, −2.34)	<0.0001	At AC_Sugar = 130	27.72 (0.31)	31.91 (0.39)	−4.19 (−5.18, −3.21)	<0.0001
At AC_Sugar = 90	26.08 (0.10)	30.24 (0.08)	−4.16 (−4.4, −3.91)	<0.0001	At AC_Sugar = 140	27.81 (0.32)	32.01 (0.40)	−4.20 (−5.21, −3.20)	<0.0001
At AC_Sugar = 100	26.59 (0.11)	31.28 (0.13)	−4.69 (−5.03, −4.36)	<0.0001	At AC_Sugar = 150	27.89 (0.32)	32.29 (0.41)	−4.40 (−5.43, −3.37)	<0.0001
At AC_Sugar = 110	27.46 (0.20)	31.72 (0.30)	−4.26 (−4.97, −3.55)	<0.0001	At AC_Sugar = 160	27.94 (0.37)	32.65 (0.48)	−4.72 (−5.91, −3.52)	<0.0001
AVI									
At AC_Sugar = 70	17.20 (0.74)	11.91 (0.45)	5.29 (3.59,6.99)	<0.0001	At AC_Sugar = 120	16.82 (0.24)	15.62(0.29)	1.20 (0.46,1.93)	0.0084
At AC_Sugar = 80	15.43 (0.21)	12.33 (0.12)	3.10 (2.62,3.58)	<0.0001	At AC_Sugar = 130	16.82 (0.27)	15.75(0.34)	1.07 (0.23,1.92)	0.0131
At AC_Sugar = 90	15.11 (0.08)	13.11 (0.07)	2.00 (1.79,2.21)	<0.0001	At AC_Sugar = 140	17.05 (0.28)	16.00 (0.34)	1.05 (0.19,1.92)	0.0172
At AC_Sugar = 100	15.64 (0.10)	14.13 (0.11)	1.52 (1.23,1.8)	<0.0001	At AC_Sugar = 150	17.43 (0.28)	16.32 (0.36)	1.11 (0.23, 2.00)	0.0139
At AC_Sugar = 110	16.46 (0.17)	15.28 (0.26)	1.18 (0.57, 1.79)	0.0009	At AC_Sugar = 160	17.87 (0.32)	16.63 (0.42)	1.24 (0.21, 2.27)	0.0181

SE: standard error. Abbreviations. DM, diabetes mellitus; AC sugar, fasting glucose; BMI, body mass index; WC, waist circumference; HC, hip circumference; WHR, waist-to-hip ratio; WHtR, waist-to-height ratio; LAP, lipid accumulation product; BRI, body roundness index; CI, conicity index; BAI, body adiposity index; AVI, abdominal volume index. Data for LAP was skewed and log transformed for analysis. Adjusted generalized linear regression model and calculated difference of least square means (Lsmean; standard error, SE) was performed for males and females with and without DM. Multiple comparison analysis testing was by using the Bonferroni method. The symbol of * means multiple by.

## Data Availability

The data underlying this study is from the Taiwan Biobank. Due to restrictions placed on the data by the Personal Information Protection Act of Taiwan, the minimal data set cannot be made publicly available. Data may be available upon request to interested researchers. Please send data requests to: Szu-Chia Chen, PhD, MD. Division of Nephrology, Department of Internal Medicine, Kaohsiung Medical University Hospital, Kaohsiung Medical University.

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
