# Peer review of "Different Curve Shapes of Fasting Glucose and Various Obesity-Related Indices by Diabetes and Sex"

_ijerph, 2021, doi:10.3390/ijerph18063096_

Round 1

Reviewer 1 Report

In this study, the relationship between fasting plasma glucose and obesity-related indices by diabetes and sex was compared. This study has a scientific contribution to the related research field. However, there are some issues to be figured out before publication. My comments are as follows:

  1. Please give full name for some abbreviations in the part of Abstract, eg. PLM, DM
  2. Please specify the novelty and significance of this study in the last paragraph in the section of Introduction.
  3. Please give the assay methods for each index in the part of 2.2 Collection of demographic, medical and laboratory data.
  4. Please give the reference for the definition of diabetes.
  5. The section of Results should be re-organized. It is better to add some subtitles in order to present the results clearly.
  6. The legends of all figures and tables should be an independent part. Please give some necessary details and add full name of some abbreviations.

Author Response

In this study, the relationship between fasting plasma glucose and obesity-related indices by diabetes and sex was compared. This study has a scientific contribution to the related research field. However, there are some issues to be figured out before publication. My comments are as follows:

  1. Please give full name for some abbreviations in the part of Abstract, eg. PLM, DM

Ans: Thank you for your kind remind. We have added the full name of PLM and DM in Abstract and Manuscript.

  1. Please specify the novelty and significance of this study in the last paragraph in the section of Introduction.

Ans: Thank you for your comments. We have added some sentences to specify the novelty and significance of this study in the last paragraph in the section of Introduction.

  • Though the sex difference of metabolism was determined in diabetes mellitus (DM) or prediabetes population [26-30], whether this difference could extend to non-DM population was uncertain, and this could affect the screening policy of related metabolic diseases in different sexes. Therefore, in this study, we enrolled 5,000 individuals from the Taiwan Biobank (TWB) database and compared curve shapes of FPG and various obesity-related indices by diabetes, and further explored sex differences in these associations. (Page 2, Introduction 3rd paragraph)

  1. Please give the assay methods for each index in the part of 2.2 Collection of demographic, medical and laboratory data.

Ans: Thank you for your comments. We have added assay methods of blood pressure and laboratory data.

  • After ten minutes rest, the blood pressures were averaged from three times measurement. Fasting blood samples were obtained, and laboratory data were measured using an autoanalyzer (Roche Diagnostics GmbH, D-68298 Mannheim COBAS Integra 400). Serum creatinine was measured according to the compensated Jaffé (kinetic alkaline picrate) method using the same autoanalyzer (Roche/Integra 400, Roche Diagnostics) and a calibrator that could be used in isotope-dilution mass spectrometry. (Page 3, first paragraph)

  1. Please give the reference for the definition of diabetes.

Ans: Thank you for your suggestion. We have added reference 37.

  1. The section of Results should be re-organized. It is better to add some subtitles in order to present the results clearly.

Ans: Thank you for your suggestion. We have added subtitles in each paragraph in Results.

3.1. Comparison of clinical characteristics of the study population between males and females with and without DM

3.2. B-spline comparisons for fasting glucose with obesity related indices

3.3. The relationship between fasting glucose and various obesity related indices by DM and sex

3.4. Separate trends between obesity related indices and fasting glucose between males and females with and without DM

  1. The legends of all figures and tables should be an independent part. Please give some necessary details and add full name of some abbreviations.

Ans: Thank you for your suggestion. We have added Abbreviations in footnotes in each table and figure legend of figure 1. Besides, we have added statistical methods in Table 1 and 3.

Reviewer 2 Report

Manuscript Number:  ijerph-1115846
Title: Different curve shapes of fasting glucose and various obesity-related indices by diabetes and sex
submitted to: IJERPH (ISSN 1660-4601)

In this article, Wei-Lun Wen et al., focuses on the study of sex differences in FPG and various obesity indices related to the risk of diabetes in adults. This study was carried out in a sample size of 5,000 individuals.

I consider that the article meets the criteria established by the journal. However, there are several points which require be clarified:

- Page 3, line 3: EGFR per eGFR

- Fig.1: Consideration should be given to increasing the size of the figure, as it is difficult to see the details of the figures.

-It is difficult to understand the first paragraph of the discussion.

-The conclusions could be improved.

Author Response

In this article, Wei-Lun Wen et al., focuses on the study of sex differences in FPG and various obesity indices related to the risk of diabetes in adults. This study was carried out in a sample size of 5,000 individuals.

I consider that the article meets the criteria established by the journal. However, there are several points which require be clarified:

  1. Page 3, line 3: EGFR per eGFR

Ans: Thank you for your corrections. We have corrected.

  1. 1: Consideration should be given to increasing the size of the figure, as it is difficult to see the details of the figures.

Ans: We have changed the style to increase the size of figure 1 to make more clear.

  1. It is difficult to understand the first paragraph of the discussion.

Ans: Thank you for your comments. We have changed the words to make more clear.

  • In this study, the different correspondences between obesity-related indices and DM status in different sexes were unmasked if we took FPG into consideration. Furthermore, the differences were more obvious in non-DM group than in the DM group but gradually declined as the FPG increased in general Taiwanese population. (Page 11, first paragraph)

  1. The conclusions could be improved.

Ans: Thank you for your comments. We have changed the sentences to make it clear.

  • In conclusion, the current study revealed a different trend of obesity-related indices in different sexes if we concerned not only DM status but added on comparing FPG. FPG affected the trend towards obesity-related indices more obviously in the non-DM group than in the DM group. Further studies with longitudinal designs would provide a better understanding of the underlying mechanisms for these relationships. (Page 12, 4th paragraph)

Round 2

Reviewer 1 Report

Thanks for your revised manuscript. I have no further comments.